# Mussel-Inspired Surface Functionalization of Porous Albumin Cryogels Supporting Synergistic Antibacterial/Antioxidant Activity and Bone-Like Apatite Formation

**DOI:** 10.3390/gels8100679

**Published:** 2022-10-20

**Authors:** Nabila Mehwish, Mengdie Xu, Muhammad Zaeem, Bae Hoon Lee

**Affiliations:** 1Engineering Research Center of Clinical Functional Materials and Diagnosis & Treatment Devices of Zhejiang Province, Wenzhou Institute, University of Chinese Academy of Sciences, Wenzhou 325011, China; 2School of Pharmaceutical Sciences, Wenzhou Medical University, Wenzhou 325000, China; 3Oujiang Laboratory (Zhejiang Lab for Regenerative Medicine, Vision and Brain Health), Wenzhou 325000, China

**Keywords:** cryogels, mussel-inspired, porous scaffolds, albumin, antibacterial, bone-like apatite

## Abstract

A crucial method for adding new functions to current biomaterials for biomedical applications has been surface functionalization via molecular design. Mussel-inspired polydopamine (PDA) has generated much attention as a facile method for the functionalization of biomaterials because of its substantial independence in deposition, beneficial cell interactions, and significant responsiveness aimed at secondary functionalization. Because of their porous structure, the bovine serum albumin methacryloyl (BSAMA)-BM cryogels were functionalized with PDA (BM-PDA), which may reproduce the architecture and biological purpose of the natural extracellular environment. Excellent antioxidative and antibacterial qualities, improved mineralization, and better cell responsiveness were all demonstrated by BM-PDA. BM-PDA scaffolds maintained their linked and uniform pores after functionalization, which can make it easier for nutrients to be transported during bone repair. As a result, hydroxyapatite (HA)-coated BM* and BM-PDA* cryogels were created through successive mineralization with the goal of mineralized bone tissue repair. The heterogeneous nucleation and surface roughness contributed to rod-like apatite production in BM-PDA* cryogels whereas BM* cryogels were made up of plate-like HA morphologies. Analysis results showed that after five cycles, the mineral contents were around 57% and the HA units remained equally dispersed on the surface of BM-PDA* with a Ca/P ratio of 1.63. Other natural polymer-based cryogels can be coated using this general, rapid, and simple PDA coating technique and utilized as implants for bone tissue engineering. Future clinical uses of albumin cryogels for bone tissue engineering will advance as a result of additional in-vivo testing of such PDA-coated cryogels.

## 1. Introduction

The mineralization of periodontal alveolar bone and teeth, where calcium phosphate crystals are deposited and developed within an extracellular matrix, is crucial to dental function and has been thoroughly proven for the establishment of a functioning skeleton [1]. Mineralization flaws in periodontal cementum and tooth dentin inevitably cause a weak dentition, which renders teeth loose, vulnerable to infection, and more likely to fall out early [2]. Despite the enormous demand for dental biomaterials, no dental materials exist that provide the optimum characteristics for any dental applications [1,3,4]. From a historical perspective, a wide range of materials, both natural and synthetic, have been applied to the treatment of numerous oral ailments. Regardless of stomatognathic system atrophy, sickness, or injury, modern dentistry aims to return the patient to normal shape, function, comfort, esthetics, speech, and health. Predictable success is now a reality for the rehabilitation of many challenging conditions as a consequence of ongoing research in treatment planning, implant designs, materials, and methodologies. A major challenge in the field of artificial tissue replacement and enhancement has always been the biocompatibility profiles of synthetic materials [5]. Moreover, these materials should mimic the appearance of a normal tooth (i.e., mineralized) and have characteristics of different dental sections [6]. Implant biomaterials must be adequate in terms of mechanical strength, biocompatibility, and structural biostability in physiologic conditions to work at their best.

Albumin, a 66.4 kDa non-glycosylated protein with a roughly 19-day physiological half-life, is prevalent in human serum [7]. It is a desired biomaterial because it stabilizes important proteins, hormones, metal ions, nanoparticles, and pharmaceuticals in addition to acting as a mild pH buffer [8,9]. Albumin hydrogels have gained increasing interest in biomedical applications (Figure 1 and Table 1), particularly in treating numerous diseases, and tissue engineering due to the excellent qualities of albumin such as sustainability, non-immunogenicity, low cost, excellent biocompatibility, and biodegradability [7,10,11,12,13,14]. Hydrogels made from albumin can be created using a photo-triggered gelation process, chemicals, heat, or pH. Cells cannot be integrated during the harsh gelation process, but the pH-induced hydrogels displayed considerable breakdown and cellular infiltration after being implanted in vivo [15]. In contrast, the heat-triggered gelation process is quick (a few minutes to 30 min) and free of harmful chemical crosslinkers. However, cells cannot survive the thermal gelation process, and albumin’s secondary structure is altered [16]. Although the photochemically crosslinked hydrogels had stable mechanical characteristics, the production technique also involved extra processes and a cost [17,18,19].

Cryogelation was recently employed to gel methacrylated BSA (BSAMA) [20] because it is a simple method for creating porous scaffolds that can protect and contain cells, as well as function as grafts in a variety of biological applications [21]. Similarly, the creation of transparent and opaque gels by using albumin methacryloyl with enhanced stability and customized porous topologies was demonstrated [22]. But according to both of these publications, cells adhered gradually over the first three days of culture and also multiplied slowly. It has been demonstrated that during the early stages of adhesion in serum-free conditions, cells exhibited better attachment and a stretched shape on GelMA than on BSAMA [23]. More than BSAMA alone, BSAMA-GelMA hybrid scaffolds seemed to be able to adapt to cells more quickly [20].

Inspired by our previous work involving albumin cryogels coating via liver ECM [24], and collagen cryogels coated with PDA [25], herein, the polydopamine (PDA) functionalization of BSAMA cryogels (BM) is carried out (Figure 1) to add improved functionalities (to increase the adherence and proliferation of cells implanted, enhance mineralization, and give antibacterial activity [26,27]) into albumin cryogel scaffolds for tissue engineering. Mussel-inspired cryogels using PDA coating can encourage mineralization, cell response, and mechanical, antibacterial, antioxidant, and anti-inflammatory qualities [28,29,30,31,32]. PDA has been frequently used to change the surface of biomaterials and is recognized as a surface modification because of its ability to bind to calcium ions [33,34,35]. Additionally, it has been demonstrated that the occurrence of the catecholamine promotes the mineralization of collagen milieu in dentine, which in turn speeds up hydroxyapatite (HA; Ca_10_(PO_4_)_6_(OH)_2_) crystallization [36]. More significantly, a single-step surface functionalization technique known as “one-pot” coating was suggested [37,38].

The ability to build a coating with various or multiple bio-functionalities was substantially increased by the discovery that synchronously injected biopolymers put into the PDA coating preserved their biological activity [25,39,40]. So, BSAMA scaffolds with PDA functionalization are designed here and are named BM-PDA (Figure 1). By using cryogelation, porous scaffolds with improved mechanical characteristics have been created. The outstanding antibacterial activities of PDA-functionalized BSAMA scaffolds aiming against both the gram-positive and gram-negative microbes will aid scaffolds in preventing infection during surgery or implantation. Furthermore, BM-PDA improved mineralization with the development of rod-like apatite and showed good antioxidative characteristics, as well as cell responsiveness. The PDA-functionalized albumin scaffolds with bone-like apatite are anticipated to not only aid in bone regeneration but also control infections that may develop after injury.

**Table 1 gels-08-00679-t001:** The summary of different preparation methods of BSA-based hydrogels and their findings.

Gelation	Findings	Limitations	Mineralization	Refs.
pH- triggered	Facile gelation of hydrogels with good biodegradable properties.	Extreme pH conditions limit the incorporation of cells.	NA	[41,42,43]
Heat-triggered	Quick, easy, and harmful chemical crosslinkers-free gelation.	Heat modifies the secondary structure. Cells cannot survive under heat-triggered gelation.	NA	[43,44,45]
Photo-triggered	Methacrylate albumin (BSAMA) hydrogels presented good cell activities and mineralization, similar to GelMA.	It entailed an additional step and expense. Incorporated cells might be damaged by UV radiation. Relatively high concentrations (15–25%) of polymer solutions were used.	On GelMA, however, there were almost twice as many mineral depositions as there were on BSAMA [23].	[23,46,47,48]
Cryogelation	A combination of urea and cysteine on BSA led to porous cryogels.	Gelation-denatured polypeptide chains.	NA	[49,50]
	BSAMA cryogels displayed a porous, interconnected spongy network with shape recovery.	Compared to BSAMA alone, the BSAMA-GelMA hybrid exhibited superior cell adhesion.	NA	[20]
	Sequentially mineralizable transparent and opaque porous gels with improved stability.	They lack multi-functionalities.	Cryogels exhibited a larger mineral percentage than their hydrogel counterparts.	[22]
	For the first time, multifunctional porous albumin gels were fabricated via surface functionalization inspired by mussels, exhibiting synergistic antibacterial/antioxidant activity and enhanced mineralization and cell responses.	In vivo investigations could be further required.	Rod-like shaped minerals, with a Ca/P ratio of 1.63 (similar to native HA 1.69) are produced by the sequential mineralization of PDA-functionalized BSAMA hydrogels.	Current study

## 2. Results and Discussion

The fractional helicity and the amide bands of the pure BSA were preserved in the BSAMA, as shown by the CD and FTIR peak data (Appendix A). To past findings [19,23], BSAMA was synthesized, and according to Appendix A, BSAMA showed the characteristic globular configuration of BSA, which raises the possibility that the two molecules share a comparable secondary structure overall. A positive peak at 193 nm and two characteristic negative bands at 212 and 222 nm are attributable to α-helicity. FTIR absorption spectra of BSA and BSAMA are displayed in Appendix A. BSAMA maintained the characteristic amide I and amide II peaks on 1651 cm^−1^ and 1546 cm^−1^, correspondingly, confirming the secondary structure similar to the native BSA [22].

Since the reports of the star coating “PDA” catechol-assisted surface chemistry has received much attention [27,28,29]. Additionally, both covalent and non-covalent interactions, such as hydrogen guideline bonding, stacking between aromatic moieties, Michael-type addition, or Schiff base reaction with amines, are essential in causing catechol to form a surface coating [31,32]. Different coating features, such as varied surface wettability, roughness, reactive group retention, antioxidant ability, etc., would be produced by altering the catechol reaction state. To examine the structure of PDA deposition on the silicon wafers, FTIR was used (Figure 2).

The main dopamine (DA) representative peaks include 3335 cm^−1^ (amine NH stretching), 3038 cm^−1^ (aromatic CH stretching), and 2961 cm^−1^ (alkyl CH stretching), as well as several absorption peaks between 650 and 1700 cm^−1^, including 1618 cm^−1^ (amine NH bending) and 1285 cm^−1^ (aromatic C=C stretching) [51]. There are three different ways that the two OH groups in dopamine can vibrate: out-of-plane bending, in-plane bending, and OH stretching [52]. Since dopamine FTIR spectra lack spectral lines in the area that are typical of OH, the scale factor of NH_2_ stretching has been employed in the literature to scale OH stretching for theoretical (Density Functional Theory (DFT)) computations. This demonstrates that the frequencies between 1180 and 1235 cm^−1^ are produced by the vibrations of the OH in-plane bending [53]. The DA small molecule can be recognized in FT-IR spectra via the strong and narrow absorption peaks, which vanish following the oxidative polymerization of DA, indicating the transformation of DA [54]. The stretching of O-H and N-H bonds and the dihydroxyindole moiety (DHI—Figure 2, upper section) produce the peaks of PDA at 3173 cm^−1^ and 1598 cm^−1^, respectively. A noticeable characteristic peak of amine NH bending in the spectra of DA is replaced by a characteristic peak of DHI in the spectrum of PDA [55]. The characteristic peaks of DA and PDA exhibit the successful oxidative polymerization of DA [56].

Spongy and porous BSAMA cryogel scaffolds (BM) were prepared and then functionalized by PDA (BM-PDA) as shown in Appendix A. Both cryogel scaffolds (BM and BM-PDA) showed sponginess; however, PDA-functionalized BM retained more water. The carboxyl groups in cryogels are thought to be able to chelate calcium ions, supplying active particle nucleating sites and triggering bio-inspired Ca/P mineralization [57]. The phosphate and choline groups interact strongly with the catechol and amino groups in the PDA structure [58]. Notably, the PDA coating layer promoted the nucleation of HA biominerals in addition to playing an anchoring role in attaching Ca^2+^ ions to the substrate [59]. Therefore, BM and BM-PDA mineralized scaffolds were prepared via sequential mineralization (Figure 3c) [60]. Sponginess decreased with an increase in mineral content from day 1 to 5, as can be seen from Appendix A.

SEM/EDX was used to observe the microstructure of cross-sections of pristine and mineralized scaffolds (Figure 3 and Figure 4). The cross-sectional morphologies of the pristine scaffolds (Figure 3a) exhibited interconnected pores. Before coating with PDA, the surface of BM pores was smooth. However, after PDA functionalization, the PDA adhered to the scaffold surface and the appearance of PDA microparticles was obvious (Figure 3a). The BM-PDA demonstrated homogeneously dispersed PDA coating. PDA seemed attached to the micro-sized apertures of BM. Compared with the micro surface of non-mineralized frameworks, the micro surface of the scaffolds after five days of mineralization became rough, HA microparticles were regularly attached to the scaffold surface, and the particles had flocculent spherical structures (Figure 3b). The SEM images of the mineralized scaffolds after day 5 (BM* and BM-PDA*) showed that PDA significantly affected the mineralization of BM in terms of mineral content and morphology. Due to the great affinity of HA for PDA moieties and the possibility of fast mineralization, the morphological alteration implies that the production of HA biominerals onto the PDA-coated surface may be based on the layer-by-layer growth paradigm. The catechol-containing PDA’s capacity to concentrate Ca^2+^ and promote CaP crystallization was likely a contributing factor In the fast”Iin’ralization [11,22].

Apatite particles that resemble flowers tightly adhered to the scaffold surface in both BM* and BM-PDA*. The biological activity (mineralization) of the BM-PDA scaffolds was boosted by increased surface area and roughness brought about by PDA. Apatite nodules from the metastable solution were also somewhat precipitated on BM-PDA scaffolds, which may have aided in the formation of rod-shaped crystals [61,62]. On the other hand, the BM substrates’ carboxyl (COOH) groups seemed to facilitate mineralization [23,63]. Therefore, heterogeneous nucleation by OH, NH, and COOH groups along with the surface roughness can be responsible for the rod morphology of bone-like apatite formed by BM-PDA.

EDX shows an increase in oxygen content for BM-PDA compared to native BM, which can be due to the functionalization of PDA with many OH groups. Furthermore, EDX confirmed the presence of calcium and phosphorus in mineralized BM* and BM-PDA* (Figure 4, Table 2). Moreover, the atomic ratio of Ca/P in BM was found to be 1.49, and that in BM-PDA was 1.63 (Table 2), indicating that PDA can provide more nucleation sites for Ca during mineralization.

Additionally, the Ca/P ratio of BM-PDA was closer to the stoichiometric Ca/P proportion of 1.67 for HA, signifying that carbonated HA was formed on the BM-PDA scaffold’s surfaces for 5 days [61]. A bone-like apatite layer was developed on the surfaces of the BM-PDA scaffolds for 5 days, according to both SEM (Figure 3) and EDX (Figure 4) analysis.

As can be seen in Figure 5a, the typical amide I and amide II peaks of BSAMA cryogels (BM) as well as an amide A band at 3270 cm^−1^ supported the original BSA’s secondary structure [19]. All samples are identical to the BSA α-helix secondary structure and have not been damaged, according to the general shape of the amide I and II bands [20,22]. Phosphate bands at 560 and 940 cm^−1^ in the FTIR spectra of the successively mineralized cryogels indicated the production of HA.

The distinctive XRD peaks (Figure 5b) showed up at angles of roughly 26° and 32°, which corresponded to the diffraction patterns (002) and (211) for HA found in the mineralized scaffolds and commercial products [22,23]. Peaks from the diffraction patterns (002), (211), (112), (300), and (004) were shared by the results for the mineralized cryogels and commercial HA between the angles of 26° and 35°. The pristine BM and BM-PDA samples were devoid of these peaks. These findings demonstrate that the sequential mineralization strategy (Figure 3c: c’) was successful in producing HA [60,64]. In addition, the effect of coated PDA on mineral deposition was observed by XRD curves. At 2θ = 26.0° and 32.0° [65], calcium phosphate deposited on BM-PDA produces relatively strong peaks, indicating that the mineral crystallinity on BM-PDA* is enhanced. This is in line with the EDX findings (Figure 4 and Table 2).

PDA makes the cryogels spongier whereas mineralization has made the scaffolds more brittle with a marked increase in mechanical strength. After mineralization, the compressive strength increased more significantly [60,66]. The increase in stress at break indicates that mineralized scaffolds have better mechanical qualities than the pristine, even if the strain has dropped (Figure 5c and Table 3).

TGA measurements were also able to demonstrate the successful preparation and mineralization of BM-PDA [57]. The moisture content of BM, BM-PDA, BM*, and BM-PDA* was estimated using the weight loss before 100 °C, as shown in Figure 5d. Between 240 and 450 °C, there was a second falling curve that could be attributed to the breakdown of BSAMA (BM). According to calculations, this period showed a weight drop of 17 wt.%. The BM-PDA experienced the third weightlessness of around 14 wt.% when the temperature was raised from 520 to 650 °C, which might lead to the disintegration of the PDA. According to this finding, PDA might easily coat BSAMA through penetration [31,67]. With a mineral content of roughly 57% and 50%, respectively, the mass loss of BM-PDA* (22.63%) was smaller than BM* (28.23%) when the temperature reached 800 °C due to the presence of PDA. Additionally, this outcome is entirely consistent with the findings of EDX and XRD as shown in Figure 4 and Figure 5b. These results suggest that PDA and HA can work together to enhance the robustness and mechanical performance of cryogels.

It is known that moisture aids in wound healing; it lessens scarring and dressing removal pain and does not obliterate newly produced tissue [68]. As a result, the produced scaffolds’ outstanding hydrophilicity and wetting qualities may be useful for wound dressing or other prospective biological applications. Herein, all scaffolds presented the sponginess and water-triggered shape recovery behavior illustrated in Appendix A and Figure 6a. It is obvious that in two seconds (Appendix A), the pristine scaffolds (BM and BM-PDA) absorbed a lot of water and showed water-triggered shape recovery, indicating that their surface had excellent hydrophilicity.

Tissue development and infiltration into biomaterial constructs are significantly influenced by the pore size, and structure. Better cell attachment and spreading are achieved on bigger pores, and interconnected pores make it easier for cells to load into scaffolds. A structure with an open network enables the flow of nutrients from the scaffold, allowing for cell survival, and proliferation. A porous structure helps to promote host cell permeation and, more significantly, the growth of vascular systems, which supply nutrients to growing tissue [69]. Therefore, the degree of pore interconnectivity (DI) was tested as shown in Figure 6b, demonstrating that pristine scaffolds have a DI value of around 75 ± 3% which is quite reasonable for cryogel scaffolds. Mineralization led to a decrease in DI to 25 ± 5%. This decrease can be attributed to the fact of mineral deposition in the pores of porous scaffolds, which is also evident in the SEM images (Figure 3). The chemical features of the material, for example, hydrophilicity, mechanical strength, and degradability are closely linked to the cell’s adherence to its surface [70]. Rapid cell attachment and high-efficiency cell seeding are made possible by the hydrophilic scaffold [71]. The comparative hydrophilicity of the prepared porous scaffolds was analyzed using a water contact angle (WCA) assay (Figure 6c). The inclusion of HA particles caused the contact angle of BM* and BM-PDA* to increase in comparison to BM and BM-PDA scaffolds, respectively whilst PDA coating significantly decreased the WCA of BM-PDA, compared to that of BM. The outcomes showed that the hydrophilicity of the biomaterials can be enhanced through hydrophilic PDA coating [72] but reduced by HA mineralization [73].

This shows that the BM scaffolds have enhanced hydrophilicity and wettability following PDA functionalization, creating an interface for quick biological reactions and biomolecule absorption. Both mineralized and pristine scaffolds had water contact angles that were less than 90 degrees, indicating that they were overall hydrophilic and potentially suitable for use as biomaterials. The swelling ability of the substance was examined for up to 24 h using three different aqueous media (DW, PBS, and growth media-GM), as shown in Figure 6d, to further explain the hydrophilicity [63]. Water-induced swelling is a sign of a substance’s hydrophilicity. PBS is a phosphate salt-based solution that maintains a pH of 7.4 and an ionic strength of 0.14 M, which are characteristics of human blood. It is common practice to analyze materials with biomedical applications using a PBS solution. The material’s behavior under physiological settings was tested at a temperature of 37 °C. The decreased mass swelling ratio of mineralized scaffolds in comparison to pristine scaffolds can be due to the relatively higher hydrophobicity of the deposited HA crystals, which may decrease pore interconnectivity and consequently lower mass swelling. The swelling ratio of BM in DW, PBS, and GM was found to be 1927%, 1617%, and 3000%, respectively in comparison to BM-PDA (2114% in DW 1727% in PBS, and 2933% in GM). There was no significant difference in the swelling ratio between BM and BM-PDA in the media.

The goal of tissue engineering is to regenerate tissue while also causing the support matrix to degrade at a rate that can keep up with the growth of new tissue [74]. During the process of tissue regeneration, the matrix molecules produced by the cells unite to form a new ECM while the scaffold deteriorates. For bone tissue engineering, a regulated scaffold biodegradation rate that is comparable to a tissue regeneration rate is preferred [75]. Therefore, proteinase K, a broad-spectrum protease whose significant proteolytic activity in native proteins can be demonstrated under mild neutral conditions, was used to test the accelerated enzymatic degradation of the as-prepared pure and mineralized scaffolds [22]. Pristine and mineralized BM and BM-PDA scaffolds exhibited different times to degrade completely (Figure 7a). It took 12 h for the complete degradation of BM whereas BM-PDA degraded 85 ± 2 % for the same period. Strikingly, mineralized BM* and BM-PDA* degraded up to 35% and 40% only over 12 h. This result can be explained by the reason that HA reinforces the overall firmness of cryogels [76], thus providing more lasting support for bone regeneration. On the other hand, all scaffolds-maintained stability (the remaining mass percentage of each sample: 80 ± 1% for BM, 83 ± 3% for BM-PDA, 85 ± 1% for BM*, and 90 ± 2% for BM-PDA*) in PBS over a period of 72 h (Figure 7b). The enzymatic degradation results demonstrate that the mineralized scaffolds decomposed more slowly than pure scaffolds. Figure 5c, Table 3 shows that adding HA to BM scaffolds enhances their mechanical characteristics and lessens the capacity of the resulting mineralized matrixes to absorb water (Figure 6d). As a result, the mineralized scaffolds deteriorated slower than the pristine ones, which is consistent with the literature report [66].

PDA functions as synthetic melanin because it can both receive and transport electrons from the reducing agent to the oxidizer [77]. PDA will quench the free radical and demonstrate free radical scavenging capabilities if the electron donor is a free radical; if the donor is oxygen, it will cause the creation of hydrogen peroxide free radicals [78]. Moreover, BSA alone has an antioxidant activity of about 30% for DPPH (a model free radical to test the antioxidant ability of any material), [79] and it is reported that BSA helps improve the antioxidant activity of polyphenols [80]. PDA has been reported to show antioxidant characteristics in cryogels [81,82]. Inflammation is probably linked to the produced free radical species [82]. Consequently, a DPPH assay was performed to confirm the free radical scavenging activity (S.A.) of the as-prepared porous spongy scaffolds (Figure 8a). For example, using 0.25 mM of DPPH as the free radical model, the BM cryogels were able to capture 33% of free radicals within 30 min. An estimated 85% of free radicals were scavenged after 30 min of incubation with PDA-functionalized cryogels (Figure 8b,c). These findings demonstrated the tremendous anti-inflammation potential of PDA-bearing cryogels.

Both Gram-positive and Gram-negative bacteria continuously adapt to their environment and develop medication resistance over time [83]. To overcome bacterial infection, the ongoing development of antibacterial substances that can successfully stop bacterial growth is vital. As a result, we used the agar plate well diffusion method to examine the antibacterial activities of BM and BM-PDA scaffolds against *S. aureus* (Gram-positive) and *E. coli* (Gram-negative) bacteria. The inhibition zones formed in a screening test after the cryogel samples were placed in the wells of an agar plate. Excellent antibacterial activity was demonstrated by the PDA-functionalized cryogels against both Gram-positive and Gram-negative bacteria as shown in Figure 8d,e. Furthermore, there was no regeneration of bacterial colonies in the inhibitory zone. However, no distinct inhibition zones were seen in the *E. coli* and *S. aureus* groups exposed to the BM cryogel after 24 h. The outcomes demonstrate that PDA coatings were the “release killing” model for antibacterial activity [30,84]. PDA ions from the cryogels may be released and stick to negatively charged bacterial cell walls via electrostatic forces, destroying the cell walls, attaching to proteins and nucleic acids of bacteria, and consequently leading to cellular deformation and the loss of viability. Additionally, it was discovered that PDA was significantly more effective against *E. coli* than *S. aureus* bacteria. This might be because the thicker Gram-positive *S. aureus* cell wall is arguably better protected against PDA particle uptake than the thinner Gram-negative cell wall [85].

By cultivating L929 fibroblasts, the interaction between the cell and scaffolds was investigated. Figure 9a show that adding PDA to the BM scaffold can promote cell adhesion and migration [86]. Despite their biological adhesion, PDAs exhibit great interactions with cell receptors (e.g., integrin) and serum proteins such as fibronectin [87]. The introduction of hydroxyl functional groups improves hydrophilicity and improves the absorption of supplementary medium to nourish cells (higher swelling behavior in growth media as exhibited in Figure 6d). The number of cells had no obvious difference for both the BM and BM-PDA on days one to five (Appendix A). However, the proliferation of cells on BM-PDA scaffolds was higher than that on BM scaffolds on day 5. In general, cells proliferated well on both scaffolds, and there was essentially no cytotoxicity in both samples. To further govern the cell response, chiral molecules can be incorporated into the cryogel system by either co-assembly or a simple surface functionalization strategy. We are planning to investigate chirality-dependent cell adhesion and differentiation for creating bone tissue and regenerative therapies in the future [88,89]. According to CLSM micrographs (Figure 9b), fibroblasts adhered well to both scaffolds (BM, BM-PDA) while the spreading of fibroblasts was more obvious in the BM-PDA scaffolds.

## 3. Conclusions

Polydopamine (PDA) was used to functionalize the bovine serum albumin methacryloyl (BSAMA) cryogels, mimicking innate ECM and biological functions thanks to their high-water content and porous structure. The scaffold’s interconnected and uniform pores might make it easier for nutrients to be transported during bone repair. Excellent antibacterial, anti-oxidative, and anti-inflammatory characteristics, as well as improved mineralization and cell responsiveness, were all demonstrated by PDA-functionalized BSAMA scaffolds. Additionally, hydroxyapatite (HA) covered scaffolds (BM* and BM-PDA*) scaffolds were prepared by sequential mineralization. The rod-like apatite formation in BM-PDA* attributed to heterogeneous nucleation and surface roughness was achieved, whereas BM scaffolds presented plate-like morphologies. The Ca/P ratio of 1.63 and the uniform distribution of the HA particles along with the 57% mineral content on the surface of the BM-PDA* was also validated by analytical data showing the similarity to the stoichiometric values for promising bone regeneration applications.

## 4. Materials and Methods

### 4.1. Materials

Deuterium oxide (D_2_O), dopamine hydrochloride, bovine serum albumin (BSA), and methacrylic anhydride (MAA, 94%) were purchased from Sigma-Aldrich (Shanghai, China). N, N, N’, N’-tetramethylethylenediamine (TEMED, 99.5%) and ammonium persulfate (APS, >98%) were purchased from Shanghai Yi en chemical technology Co., Ltd. (Shanghai, China). Penicillin/streptomycin (P/S), fetal bovine serum (FBS), Trypsin-EDTA, PBS, Dulbecco’s modified Eagle’s medium (DMEM), and live/dead cell viability/cytotoxicity kit were provided by Thermo Fisher Scientific Inc. (Waltham, MA, USA). We obtained a ready-to-use DAPI staining solution from Phygene Life Sciences Co., Ltd. (Fuzhou, China). We bought Fluorescence Mounting Medium from Agilent Technologies (Glostrup, Denmark).

### 4.2. Synthesis of BSAMA

BSAMA (DS-100) was synthesized by already reported methods [19,20,23]. The clear solution of 5% (to get a porous scaffold as the lower the polymer concentration, the bigger the pores) BSAMA in DI water, named BM, was stored at 4 °C until further use. The structural integrity of BSAMA in comparison to BSA was evaluated by FTIR and CD.

### 4.3. Scaffold Preparation

#### 4.3.1. Cryogelation

The previously stated method of dissolving the precursor solution in the presence of an initiator and catalyst, followed by freezing and thawing, was used to make all of the cryogels. Herein, 5% BSAMA subjected to DI water was incubated at 4 °C for complete dissolution. The obtained clear solution of 5% BSAMA was subjected to 0.5% of a 10% APS solution followed by the addition of 0.25% TEMED over an ice bath. After gentle mixing, 110 µL of the precursor solution was loaded onto each section of the silicone rubber mold and incubated at −20 °C overnight [20]. The as-prepared BM cryogels were stored at 4 °C or lyophilized until further tests.

#### 4.3.2. Functionalization by PDA Coating

Dopamine hydrochloride (DA) solution (5 mg mL^−1^) was prepared in PBS (pH~8.5), and the solution pH was maintained by using 1N NaOH. Each BM cryogel was subjected to the DA solution and was incubated at 37 °C overnight to get an adequate PDA functionalization. Before further testing, PDA-functionalized BM cryogels were lyophilized or kept at 4 °C after being rinsed with DI water to eliminate any unattached PDA. The silicon wafer was chosen as the model substrate to further assess the coatings’ post-deposition properties because of its diverse adhesive ability on a variety of substrates.

#### 4.3.3. Mineralization

Mineralization for prepared scaffolds was conducted according to the already reported method [1,60]. Calcium chloride (CaCl_2_) and disodium hydrogen phosphate (Na_2_HPO_4_) solutions were produced in 0.1 M Tris buffer (pH = 7.4) at concentrations of 400 × 10^−3^ M and 240 × 10^−3^ M, respectively, and were autoclaved for sterilization. Each cryogel disk underwent one cycle treatment, consisting of a 12-h incubation in the CaCl_2_ solution, followed by a 12-h incubation in the Na_2_HPO_4_ solution, and a final 12-h rinse in deionized water (or one-day treatment). After five days, mineralized cryogels were produced. To confirm the distribution of the Ca and P elements in the samples and the element ratio between calcium and phosphorus (Ca/P), the pristine (without mineralization, day 0) and mineralized cryogels were lyophilized and stored until further testing.

### 4.4. Morphology by SEM and EDX

The Field Emission Scanning Electron Microscopy (FE-SEM-SU8020, Hitachi, Japan) was used for morphology experiments at a voltage of 5 KV. Representative cryogel samples were lyophilized after being first frozen at −80 °C. In a high vacuum sputter coater, a thin layer of platinum was applied to the samples before testing (LEICA, Wetzlar, Germany, EM ACE600). Furthermore, to confirm the distribution of the Ca and P elements in the samples and the element ratio between calcium and phosphorus (Ca/P) by SEM-coupled EDX, the pristine (without functionalization and mineralization) and PDA-functionalized and mineralized scaffolds (obtained after five days) lyophilized scaffolds were used.

### 4.5. FTIR/CD

Using a Vertex 70 FTIR spectrometer (Bruker, Billerica, MA, USA) outfitted with a mercury cadmium telluride photodetector (Bruker) and a MIRacle ATR accessory module with a three-reflection ZnSe ATR crystal, ATR-FTIR spectra of BSA and BSAMA precursor solutions (10 mg mL^−1^) were acquired in triplicate (PIKE Technologies, Fitchburg, WI, USA). Averaging 16 scans, spectra were collected from 4000 to 1000 cm^−1^ at a resolution of 4 cm^−1^. The spectra of the blank air and water vapor were subtracted from those of each sample to eliminate the water vapor contribution from the acquired spectra. The same process was used for scaffolds (thin films made by using Mini Hydraulic Pellet Press (GS01152 Mini Pellet Press Kit Asia, New Taipei City, Taiwan) for 7 mm Pellets.

To better understand the secondary structure of BSA and BSAMA, circular dichroism (CD) studies were carried out using Chirascan Plus (Applied Photophysics, Leatherhead, UK). DI water was used to dilute 10% BSA and BSAMA to a concentration of 0.1 mg mL^−1^. Each solution was kept at 4 °C for two hours before measurement to provide a steady signal of each protein. Then, a quartz cell with an optical path length of 1 mm was examined with 300 µL of each solution poured inside.

### 4.6. Mechanical Properties

To examine the mechanical qualities of the cryogels, compression tests at 95% strain were performed. Using mechanical testing equipment, the prepared cylinder triplicate samples of each cryogel scaffold were squeezed at a rate of 1 mm min^−1^ under equilibrium swelling circumstances (UTM2102; Shenzhen, China).

### 4.7. TGA

By measuring the weight of the sample as a function of temperature, the Perkin Elmer Pyris Dimond model TGA 4000 (Waltham, MA, USA) was used to determine the cryogels’ thermal stability (ASTM E1131). TGA results were obtained after heating 10 mg of each sample (pure and mineralized, ground into powder after lyophilization) at a heating rate of 10 °C per minute in the temperature range of 30–800 °C in a nitrogen environment (N_2_ flow: 50 cm^3^ per minute) [90].

### 4.8. XRD

Pristine and mineralized lyophilized samples were crushed into powder for X-ray diffraction (XRD) analysis. XRD tests were carried out with the Bruker D8 Advance (Malvern Panalytical, Westborough, MA, USA). A nickel filter, 1/2-inch diverging slit, vertical goniometer, and X-Celerator detector were employed with the apparatus. The measurements were taken under CuKα (λ = 1.542) at 2θ = 5–60°. For the aforementioned tests, three duplicates were employed, and pure commercial HA served as a control. The International Centre for Diffraction Data’s database for XRD analyses was then used to examine the patterns.

### 4.9. Swelling Behavior

The lyophilized cryogels were weighted (W_d_) and submerged in three different solvents (DW, PBS, and growth medium) at various time intervals. Swollen samples were weighted again (W_i_) at each interval until they reached equilibrium swelling. The absorbed water mass was divided by W_d_ to produce the swelling ratio by Equation (1).
Swelling Ratio (%) = (W_i_ − W_d_)/W_d_ × 100%(1)

A similar procedure was followed for mineralized scaffolds.

### 4.10. Degree of Pore Interconnectivity (DI)

Using the water-wicking technique, the degree of pore interconnectivity (DI) of the mineralized and pristine scaffolds was assessed. The interconnected porosity was determined as the interconnected void volume over the total volume. Cryogels were weighed after 1 h of water soaking to establish total volume (W_s_). Following the application of a Kim wipe to remove any remaining free water from linked pores, the cryogels were once again weighed (W_w_). According to Equation (2), the values were utilized to determine the degree of pore interconnectivity.
DI (%) = (W_S_ − W_w_)/W_S_ × 100%(2)

### 4.11. Water Contact Angle (WCA)

Attention Theta Optical Tensiometer was used to measure the water contact angle (WCA) of the pristine and mineralized scaffolds at room temperature, and OneAttention software was used to image the results to show the degree of hydrophilicity [91].

### 4.12. Enzymatic Degradation

Cryogels in a disc form were submerged in PBS until they reached the equilibrium swelling. Each cryogel had the liquid on its surface gently blotted off using KimWipes before its initial mass was weighed (W_0_). Each disc-shaped gel was broken down at 37 °C in a 2 mL PBS solution with 0.1 mg mL^−1^ Proteinase K. Every day, the enzyme solution was changed. Each cryogel’s surface liquid was removed at predetermined intervals (1, 3, 5, 7, 9, 12 h, etc.), and the remaining mass was weighed (W_t_). Using Equation (3), the percentage of degradation was used to determine each gel’s biodegradation property, and three parallel samples were examined.
Degradation (%) = (W_0_ − W_t_)/W_0_ × 100%(3)

To check the stability of the cryogels, each sample was subjected to 2 mL PBS and the weight change was recorded following a similar procedure.

### 4.13. Antibacterial Properties

An agar diffusion test was used to assess the scaffold’s antimicrobial properties against *S. aureus* and *E. coli*. First, bacteria were suspended at a final concentration of 10^5^–10^6^ CFU mL^−1^ in a bacterium buffer with a pH of 7. Second, 100 µL of the bacterial suspension was introduced to the cryogel scaffolds after they had already been made. The Zone of Inhibition (ZOI) test was used to qualitatively evaluate the antibacterial capabilities of scaffolds. A fresh agar plate that had been seeded with 100 µL of the bacterial suspension (10^6^ CFU mL^−1^) was used for the ZOI test. Each cryogel disc was sterilized in an autoclave before being placed on a new agar plate containing a bacterial suspension. To check for ZOI, the plates were incubated at 37 °C for 24 h.

### 4.14. Antioxidant Activity of Scaffolds

According to published methodologies, the potential antioxidant activity of as-prepared cryogels was assessed using a 2,2-diphenyl-1-picrylhydrazyl (DPPH) radical-scavenging assay [82,92]. Due to its insolubility in water, 0.1 mM of DPPH in ethanol was freshly produced before usage. The sample was incubated with a certain volume of the DPPH/ethanol solution for some time in the dark. Using a spectrophotometer, the UV absorbance of DPPH at 516 nm was measured (UV-2550; Shimadzu, Kyoto, Japan). Additionally, the following equation was used to compute DPPH-radical scavenging activity (DPPH S.A.).
DPPH S.A. (%) = (A_c_ − A_s_)/A_c_ × 100%(4)
where A_c_ is the absorbance of a pure DPPH solution in the absence of samples, and A_s_ is the absorbance of the sample mixed with DPPH.

### 4.15. Biocompatibility

#### 4.15.1. Cell Seeding

Cell seeding mouse fibroblast cells (L929) were cultured on sterile cryogels by the ISO 10993 biocompatibility testing standard for dental repair biomaterials, which was approved by the FDA [93]. A 48-well plate has one well for each sample. Before cell seeding, cryogels were mechanically compressed to remove part of the free water. A cell suspension was applied to each sterile scaffold (10,000 cells mL^−1^). The cell suspension was evenly spread across the top of the scaffold using the dropwise approach. First, 2 h at 37 °C and 5% CO_2_ were spent incubating cell-seeded scaffolds. Then, a 1 mL addition of enhanced DMEM was made gradually and carefully. Up to five days were spent growing the constructs, with media being changed every other day.

#### 4.15.2. Live/Dead Assay

The cytotoxicity of the resultant scaffolds was evaluated via live/dead staining. Live cells were labeled with 1.5 µM calcein-acetomethoxy (calcein-AM) and dead cells with 3 µM ethidium homodimer-1 (EthD-1). After spending 30 min in the incubator, samples were gently washed with PBS. Using a confocal laser scanning microscope (CLSM), cells were examined and captured on camera (Nikon, A1, Tokyo, Japan).

#### 4.15.3. Cell Morphology

On day five, samples were fixed for 15 min in 4% paraformaldehyde before being thrice washed in PBS for fluorescence staining. Following that, the cells were permeabilized in PBS containing 0.1% Triton X-100 and 1% BSA for 30 min at 4 °C. After that, seeded cells were stained with phalloidin for 40 min to visualize the actin cytoskeletons and ready-to-use DAPI solution for 5 min to describe the nuclei. Cells were examined and captured using CLSM.

### 4.16. Statistical Analysis

GraphPad prism5 and OriginPro 8.5.1 were used to analyze the data. Triplicate samples were used for each condition, unless otherwise specified, and data were given as mean standard deviation (SD). The two-tailed *t*-test was used to determine the statistical significance of the results after the data were checked for normality to access the normal distribution. In statistical comparisons, *p* values lower than 0.05 were regarded as statistically significant.

## Data Availability

The data generated from the study are presented and discussed in the manuscript.

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
