# Peer review of "Mussel-Inspired Surface Functionalization of Porous Albumin Cryogels Supporting Synergistic Antibacterial/Antioxidant Activity and Bone-Like Apatite Formation"

_gels, 2022, doi:10.3390/gels8100679_

Round 1
Reviewer 1 Report
Dear authors,
Here are my comments:
1. Which are the novelty and added value of the paper with respect to existent literature?
2. Have you tried SBF Kokubo method for mineralization? A comparison could be interesting?
3. Standard for mechanical properties should be added.
4. Rheological measurements are important for gels. G' and G'' could be added.
5. Have you checked the swelling kinetics (Fickian, chain-controlled, relaxation etc.)?
6. Have you determined the free surface energies together with contact angles?
Author Response
Reviewer 1
Here are my comments:
We are grateful for the chance to revise the work. Here is our point-by-point response to your comments in bold (bold).
- Which are the novelty and added value of the paper with respect to existent literature?
Response: We have provided a table (Table 1) in the revised manuscript to highlight the novelty and added value of the paper concerning existing literature.
Table 1 The summary of different preparation methods of BSA-based hydrogels and their findings.
|
Gelation |
Findings |
Limitations |
Mineralization |
Ref. |
|
pH- triggered |
Facile gelation of hydrogels with good biodegradable properties. |
Extreme pH conditions limit the incorporation of cells. |
NA |
[1-3] |
|
Heat-triggered |
Quick, easy, and harmful chemical crosslinkers-free gelation. |
Heat modifies the secondary structure. Cells cannot survive under heat-triggered gelation. |
NA |
[3-6] |
|
Photo-triggered |
Methacrylate albumin (BSAMA) hydrogels presented good cell activities and mineralization, similar to GelMA. |
It entailed an additional step and expense. Incorporated cells might be damaged by UV radiation. Relatively high concentrations (15-25%) of polymer solutions were used. |
BSAMA hydrogels offered a higher quality of mineralization than GelMA.[7] |
[7-12] |
|
Cryogelation
|
A combination of urea and cysteine on BSA led to porous cryogels. |
Gelation-denatured polypeptide chains. |
NA |
[13,14] |
|
BSAMA cryogels displayed a porous, interconnected spongy network with shape recovery. |
Compared to BSAMA alone, the BSAMA-GelMA hybrid exhibited superior cell adhesion. |
NA |
[15] |
|
|
Sequentially mineralizable transparent and opaque porous gels with improved stability. |
They lack multi-functionalities. |
Cryogels exhibited a larger mineral percentage than their hydrogel counterparts. |
[16] |
|
|
For the first time, multifunctional porous albumin gels were fabricated via surface functionalization inspired by mussels, exhibiting synergistic antibacterial/antioxidant activity and enhanced mineralization and cell responses. |
In vivo investigations could be further required. |
Rod-like shaped minerals with a Ca/P ratio of 1.63 (similar to native HA 1.69) were produced by the sequential mineralization of PDA-functionalized BSAMA hydrogels. |
Current study |
- Have you tried SBF Kokubo method for mineralization? A comparison could be interesting?
Response: We agree with the reviewer about the interesting comparison and findings of using the SBF method for mineralization. To improve the bonding capacity of titanium (Ti) metal implants to the bone, CaP coatings were originally explored in the early 1980s. Since then, several approaches have been developed employing coating techniques such as thermal spraying, sputter coating, dip coating, SBF, and sequential mineralization to impart bioactivity to non-bioactive materials. There are benefits and drawbacks to each of these strategies.[17,18] For example, although SBF coating may produce bone-like apatite, it requires lengthy processing periods, resupply is required to maintain the ionic concentration and pH of SBF, and coating is restricted by certain thicknesses. On the other hand, SBF mineralization could take more time than sequential mineralization (data not shown). Comparatively, sequential mineralization has shown to be an effective technique for mineralizing 3D porous scaffolds in a manageable and quick way. Therefore, in the current investigation, the mineralization of the cryogel scaffolds was done sequentially.
- Standard for mechanical properties should be added.
Response: We are highly thankful to the reviewer for bringing this to our notice; we have revised the mechanical test figure and also have summarized a table (Table 3) for clear understanding.
Table 3. Mechanical properties of BM and BM-PDA pristine and mineralized cryogels.
|
Sample |
Stress at Break (kPa) |
Strain at Break (%) |
Compressive Modulus (kPa) |
|
BM |
6.00 ± 7.35 |
84.56 ± 1.03 |
0.26787 ± 0.01217 |
|
BM* |
44.00 ± 0.20 |
85.74 ± 2.37 |
2.06765 ± 0.06343 |
|
BM-PDA |
3.00 ± 4.72 |
95.70 ± 0.16 |
0.07683 ± 0.00332 |
|
BM-PDA* |
40.00 ±. 0.80 |
71.43 ± 1.23 |
2.42127 ± 0.07817 |
- Rheological measurements are important for gels. G' and G'' could be added.
Response: In our earlier investigation, we tested the rheological behavior of BSAMA cryogels in considerable detail.[19] We decided not to provide the rheological measurements here to minimize repetitions. In addition, for polymeric materials for which Hook's law is true, there is a straightforward correlation between Young's modulus E and the shear modulus G (Equation 1).[20,21]
E ≈ 3 G Equation 1
- Have you checked the swelling kinetics (Fickian, chain-controlled, relaxation etc.)?
Response: Water can easily enter the pores and the interior of the pore wall (polymer) regions of the gels because of their macroporous structure and hydrophilic nature, which effectively swells the cryogels.[22] The cryogel pore wall thicknesses and cross-linking levels have been found to affect the equilibrium swelling ratios of the cryogels; lower wall thicknesses and lower cross-linking levels result in larger swelling ratios because the flexible pores make network expansion easier. Whereas, Fick's laws generally apply to one-dimensional films, particle gas transport and diffusion in semiconductor systems, and physically cross-linked polymers.[23] Alternatively, chain relaxation plays a crucial role in the shrinkage of elastomeric foams. In our work, we didn't examine the swelling kinetics (Fickian, chain-controlled, relaxation, etc.), which are typically made for drug delivery systems; rather, the swelling kinetics of cryogels was carried out by literature by using a weighing technique to follow the dynamic swelling behavior.[25,26]
- Have you determined the free surface energies together with contact angles?
Response: Data that indicate the thermodynamics of a liquid-solid interaction are obtained through measurements of contact angles on solids. Drops of various liquids (hydrophilic and hydrophobic) are formed on a solid surface to calculate surface tension via contact angle measurement. These data are used to calculate the chemical properties of the solid surface such as critical surface tension, surface tension, surface free energy, wettability.[27] The scope of our reported work does not permit such intricate calculations; the simple hydrophilicity of the hydrogels was considered when the water contact angles were examined.
- Yuan, H.; Zheng, X.; Liu, W.; Zhang, H.; Shao, J.; Yao, J.; Mao, C.; Hui, J.; Fan, D. A novel bovine serum albumin and sodium alginate hydrogel scaffold doped with hydroxyapatite nanowires for cartilage defects repair. Colloids Surf B Biointerfaces 2020, 192, 111041, doi:10.1016/j.colsurfb.2020.111041.
- Hájovská, P.; Chytil, M.; Kalina, M. Rheological study of albumin and hyaluronan-albumin hydrogels: Effect of concentration, ionic strength, pH and molecular weight. Int J Biol Macromol 2020, 161, 738-745, doi:10.1016/j.ijbiomac.2020.06.063.
- Arabi, S.H.; Aghelnejad, B.; Schwieger, C.; Meister, A.; Kerth, A.; Hinderberger, D. Serum albumin hydrogels in broad pH and temperature ranges: characterization of their self-assembled structures and nanoscopic and macroscopic properties. Biomater Sci 2018, 6, 478-492, doi:10.1039/c7bm00820a.
- Khanna, S.; Singh, A.K.; Behera, S.P.; Gupta, S. Thermoresponsive BSA hydrogels with phase tunability. Mater Sci Eng C Mater Biol Appl 2021, 119, 111590, doi:10.1016/j.msec.2020.111590.
- Bercea, M. Self-Healing Behavior of Polymer/Protein Hybrid Hydrogels. Polymers (Basel) 2021, 14, doi:10.3390/polym14010130.
- Ong, J.; Zhao, J.; Levy, G.K.; Macdonald, J.; Justin, A.W.; Markaki, A.E. Functionalisation of a heat-derived and bio-inert albumin hydrogel with extracellular matrix by air plasma treatment. Sci Rep 2020, 10, 12429, doi:10.1038/s41598-020-69301-7.
- Chen, Y.; Zhai, M.J.; Mehwish, N.; Xu, M.D.; Wang, Y.; Gong, Y.X.; Ren, M.M.; Deng, H.; Lee, B.H. Comparison of globular albumin methacryloyl and random-coil gelatin methacryloyl: Preparation, hydrogel properties, cell behaviors, and mineralization. International Journal of Biological Macromolecules 2022, 204, 692-708, doi:https://doi.org/10.1016/j.ijbiomac.2022.02.028.
- Mao, S.-Y.; Peng, H.-W.; Wei, S.-Y.; Chen, C.-S.; Chen, Y.-C. Dynamically and Spatially Controllable Albumin-Based Hydrogels for the Prevention of Postoperative Adhesion. ACS Biomaterials Science & Engineering 2021, 7, 3293-3305, doi:10.1021/acsbiomaterials.1c00363.
- Yoon, H.; Lee, H.; Shin, S.Y.; Jodat, Y.A.; Jhun, H.; Lim, W.; Seo, J.W.; Kim, G.; Mun, J.Y.; Zhang, K.; et al. Photo-Cross-Linkable Human Albumin Colloidal Gels Facilitate In Vivo Vascular Integration for Regenerative Medicine. ACS Omega 2021, 6, 33511-33522, doi:10.1021/acsomega.1c04292.
- Smith, P.T.; Altin, G.; Millik, S.C.; Narupai, B.; Sietz, C.; Park, J.O.; Nelson, A. Methacrylated Bovine Serum Albumin and Tannic Acid Composite Materials for Three-Dimensional Printing Tough and Mechanically Functional Parts. ACS Appl Mater Interfaces 2022, 14, 21418-21425, doi:10.1021/acsami.2c01446.
- Lantigua, D.; Nguyen, M.A.; Wu, X.; Suvarnapathaki, S.; Kwon, S.; Gavin, W.; Camci-Unal, G. Synthesis and characterization of photocrosslinkable albumin-based hydrogels for biomedical applications. Soft Matter 2020, 16, 9242-9252, doi:10.1039/d0sm00977f.
- Ferracci, G.; Zhu, M.; Ibrahim, M.S.; Ma, G.; Fan, T.F.; Lee, B.H.; Cho, N.J. Photocurable Albumin Methacryloyl Hydrogels as a Versatile Platform for Tissue Engineering. ACS Appl Bio Mater 2020, 3, 920-934, doi:10.1021/acsabm.9b00984.
- Rodionov, I.A.; Grinberg, N.V.; Burova, T.V.; Grinberg, V.Y.; Lozinsky, V.I. Cryostructuring of polymer systems. Proteinaceous wide-pore cryogels generated by the action of denaturant/reductant mixtures on bovine serum albumin in moderately frozen aqueous media. Soft Matter 2015, 11, 4921-4931, doi:10.1039/C4SM02814G.
- Lozinsky, V.I.; Shchekoltsova, A.O.; Sinitskaya, E.S.; Vernaya, O.I.; Nuzhdina, A.V.; Bakeeva, I.V.; Ezernitskaya, M.G.; Semenov, A.M.; Shabatina, T.I.; Melnikov, M.Y. Influence of succinylation of a wide-pore albumin cryogels on their properties, structure, biodegradability, and release dynamics of dioxidine loaded in such spongy carriers. Int J Biol Macromol 2020, 160, 583-592, doi:10.1016/j.ijbiomac.2020.05.251.
- Mehwish, N.; Chen, Y.; Zaeem, M.; Wang, Y.; Lee, B.H.; Deng, H. Novel biohybrid spongy scaffolds for fabrication of suturable intraoral graft substitutes. International Journal of Biological Macromolecules 2022, 214, 617-631, doi:https://doi.org/10.1016/j.ijbiomac.2022.06.125.
- Xu, M.; Mehwish, N.; Lee, B.H. Facile Fabrication of Transparent and Opaque Albumin Methacryloyl Gels with Highly Improved Mechanical Properties and Controlled Pore Structures. Gels 2022, 8, 367.
- Wu, X.; Walsh, K.; Hoff, B.L.; Camci-Unal, G. Mineralization of Biomaterials for Bone Tissue Engineering. Bioengineering (Basel) 2020, 7, doi:10.3390/bioengineering7040132.
- Shin, K.; Acri, T.; Geary, S.; Salem, A.K. Biomimetic Mineralization of Biomaterials Using Simulated Body Fluids for Bone Tissue Engineering and Regenerative Medicine<sup/>. Tissue Eng Part A 2017, 23, 1169-1180, doi:10.1089/ten.TEA.2016.0556.
- Mehwish, N.; Chen, Y.; Zaeem, M.; Wang, Y.; Lee, B.H.; Deng, H. Novel biohybrid spongy scaffolds for fabrication of suturable intraoral graft substitutes. Int J Biol Macromol 2022, 214, 617-631, doi:10.1016/j.ijbiomac.2022.06.125.
- Mott, P.H.; Roland, C.M. Limits to Poisson's ratio in isotropic materials. Physical Review B 2009, 80, 132104, doi:10.1103/PhysRevB.80.132104.
- Relationship Between Moduli of Elasticity. Available online: https://polymerdatabase.com/polymer%20physics/Moduli.html (accessed on
- Shirbin, S.J.; Karimi, F.; Chan, N.J.; Heath, D.E.; Qiao, G.G. Macroporous Hydrogels Composed Entirely of Synthetic Polypeptides: Biocompatible and Enzyme Biodegradable 3D Cellular Scaffolds. Biomacromolecules 2016, 17, 2981-2991, doi:10.1021/acs.biomac.6b00817.
- Devan, K.; Bachchan, A. Chapter 4 - Homogeneous reactor and neutron diffusion equation. In Physics of Nuclear Reactors, Mohanakrishnan, P., Singh, O.P., Umasankari, K., Eds.; Academic Press: 2021; pp. 193-262.
- Schott, H. Swelling kinetics of polymers. Journal of Macromolecular Science, Part B 1992, 31, 1-9, doi:10.1080/00222349208215453.
- Gun'ko, V.M.; Savina, I.N.; Mikhalovsky, S.V. Cryogels: morphological, structural and adsorption characterisation. Adv Colloid Interface Sci 2013, 187-188, 1-46, doi:10.1016/j.cis.2012.11.001.
- Caló, E.; Barros, J.; Ballamy, L.; Khutoryanskiy, V.V. Poly(vinyl alcohol)–Gantrez® AN cryogels for wound care applications. RSC Advances 2016, 6, 105487-105494, doi:10.1039/C6RA24573K.
- Gindl, M.; Sinn, G.; Gindl, W.; Reiterer, A.; Tschegg, S. A comparison of different methods to calculate the surface free energy of wood using contact angle measurements. Colloids and Surfaces A: Physicochemical and Engineering Aspects 2001, 181, 279-287, doi:https://doi.org/10.1016/S0927-7757(00)00795-0.

Reviewer 2 Report
The current work needs minor revision before publication in Gels journal. Critical remarks are listed below:
Authors should check for typos and grammatical errors in the manuscript.
Authors can be consistent with the terms gel, hydrogel or cryogel, throughout the work, according to the correct definition;
Lines 117, 118: Please check the sentence;
Lines 138-142: Please check and correct the assigning of the wavenumber values to the bond's vibrations. Please identify the O-H vibration in the spectra;
Line 144: Please check and replace the inappropriate term used “decomposition” with transformation, for example;
Line 152: Please check and replace the inappropriate term used “behavior”, because it's about how it looks;
Line 156: In fig. 2., the values of the scale bar are inconclusive;
Line 231: Please check and correct the expression “greater mechanical qualities”, a comparison of the values with those of other materials would be required;
Lines 286, 287: Please revise the explanation regarding the decrease of swelling after the mineralization of BM and BM-PDA;
Line 307: An explanation is required ;
Line 342: Please check the sentence;
Lines 412, 413, 437: Please verify the sentences;
Line 460: Please check and correct the measure unit;
Line 474: Please check the sentence;
Line 508: Please check the correctness of the equation (3);
Line 516: Please check the sentence.
Author Response
Reviewer 2
The current work needs minor revision before publication in Gels journal. Critical remarks are listed below:
We appreciate having the opportunity to edit the work. We have thoroughly revised the original content in response to your critical remarks; we have proofread it several times to reduce grammatical and typographical problems. Here, we present the answers to the reviewers' comments (in italics language) (in bold). In addition, the article has been edited in response to the reviewers' suggestions, and the updated version of the manuscript highlights the modifications.
Authors should check for typos and grammatical errors in the manuscript.
Response: We have carefully proofread the manuscript to minimize typographical and grammatical errors
Authors can be consistent with the terms gel, hydrogel or cryogel, throughout the work, according to the correct definition;
Response: We appreciate the reviewer’s comment about the consistency of the term used in this manuscript. Our paper has emphasized the different kinds of gels produced using albumin, so we can not use one term in the introduction. According to your suggestion, we have revised the discussion part with the term cryogel used throughout since we used the cryogelation strategy to make gels.
Lines 117, 118: Please check the sentence;.
Response: We have checked and corrected the sentence.
Lines 138-142: Please check and correct the assigning of the wavenumber values to the bond's vibrations. Please identify the O-H vibration in the spectra;
Response: Thank you for your comments. We have checked and corrected the wavenumber values to the bond's vibrations. Moreover, the O-H vibration in the spectra has been identified according to the literature.
Line 144: Please check and replace the inappropriate term used “decomposition” with transformation, for example;
Response: As advised, we have checked and replaced the term with transformation.
Line 152: Please check and replace the inappropriate term used “behavior”, because it's about how it looks;
Response: We have checked and replaced the term.
Line 156: In fig. 2., the values of the scale bar are inconclusive;
Response: Thank you for your comment. We have revised the values of the scale bar of Figure 2(Figure 3 in the revised manuscript).
Line 231: Please check and correct the expression “greater mechanical qualities”, a comparison of the values with those of other materials would be required;
Response: Thank you for your comment. We have checked and corrected the expression.
Lines 286, 287: Please revise the explanation regarding the decrease of swelling after the mineralization of BM and BM-PDA;
Response: Thank you for your comment. We have checked and revised the explanation.
Line 307: An explanation is required ;
Response: Thank you for your comment. We have provided the explanation according to your comment.
Line 342: Please check the sentence;
Response: We have checked and removed the sentence for better clearance.
Lines 412, 413, 437: Please verify the sentences;
Response: We have checked and corrected sentences.
Line 460: Please check and correct the measure unit;
Response: We have checked and corrected unit.
Line 474: Please check the sentence;
Response: We have checked and corrected the sentence.
Line 508: Please check the correctness of the equation (3);
Response: We have checked and corrected equation (3).
Line 516: Please check the sentence.
Response: We have checked the sentence, and it is correct.

Reviewer 3 Report
The authors have submitted an interesting article " Mussel-inspired surface functionalization of porous albumin gels supporting synergistic antibacterial/antioxidant activity and bone-like apatite formation" which deals with the implementation of mussel-inspired chemistry to boost the antimicrobial property of porous albumin gels, as well as promoting mineralization of in bone. The manuscript is well structured and reads well overall, although it will need some revisions. I suggest this article be published after a major revision.
*** General comments:
ü The abstract is clear and comprehensive, and it comprises all cornerstones including a brief/general introduction to the topic, a non-technical summary of the major findings, and their implications.
ü The introduction is compelling, clear, and concise. The introduction part covers a proper description of the challenge/gap and a strong background in the field associated with a fair literature review.
ü The various sections of the body of the text are clear and concise overall.
ü The conclusions are logical.
*** Suggested revisions:
1- First of all, I strongly recommend the authors provide a simple, high-quality, and informative “Graphical abstract” which can present the whole concept of your study at a glance. I would like to recommend authors design a “Graphical Abstract” for this study to better show the whole story in a simple and informative manner. In this regard, you can use illustrate a simple sketch of the big picture and add elements like Schematic1 along with some SEM images along with Live/Dead and a simple schematic of grafting maybe (totally up to you)
2- Please carefully revise the manuscript to remove grammatical errors and vague sentences. Some of the sentences are unnecessarily long which makes it difficult and boring for the readers to follow them. Please double-check the whole manuscript and revise all.
* I strongly suggest the author keep consistency in the figure's presenting style. For example, FTIR spectra presented in the main body of the manuscript is in Transmittance but supplementary material is presented based on the absorbance rate!
* Moreover, the scale bare presenting style in figure 3 is different from the rest of the manuscript.
* Furthermore, it is better to be consistent in the way of labeling section of the figure (from left to the right or top to bottom – see figure 4 vs figure 5). Besides, Please decrease the font size of writings in some figures such as figures 6, 7, etc.
* I suggest presenting the mass loss profile in an inverted manner (Remaining mass(%)) which is a more intuitive way to present this type of data ( a negative slope - decreasing from 100% ).
3- The novelty statement of an article is of significant importance that highlights the importance of the current study and separates it from previously done research. In this work, the novelty statement poorly represents the work and the authors needed more development and better define their hypothesis and objectives and how the presented work differs from already and recently published reports in the field with similar concepts.
4- One of the best ways to highlight the outcomes of a study is to tabulate a comparative master table to compare the findings of this study with recently published data from other research groups. I suggest the authors add a such table to the introduction part.
5- Figure 8. a doesn’t represent proper Live/dead assay data. To avoid confusion, please assign the live images and dead images with green “Live” and red “Dead” at the top left of the images and also provide merged images.
6- In the supplementary information, the FTIR spectrum should be revised to Y-axis: Absorbance (a.u.). Same comment for Figure 7.b, and figure 4.a (transmittance (%)) and figure 4.b (Intensity (a.u.)).
7- Since the SEM technique is associated with repetitive vacuum conditions (Au/or Pt coating and image-taking steps) which can dry out the microorganism and may change their original shape, therefore, authors are required to explain how they have prepared samples for SEM while preserving the original morphology of these microorganisms.
8- Some of the references in the context are too old (e.g., 1994, 2003, etc.) and it is not acceptable at all. A myriad of research bodies has been published in recent years and you can find similar concepts and cite them in your paper rather than more than 2 or 3 decades old references. Moreover, in the introduction part to better explain the fundamentals of this field, please read and add valuable information from the following key papers as well:
Introduction section/ lines 91: “Mussel‐Inspired Biomaterials: From Chemistry to Clinic - https://doi.org/10.1002/btm2.10385 “
Introduction section/ lines 94: “Mussel-inspired hydrogels: from design principles to promising applications- https://doi.org/10.1039/C9CS00849G ”
Introduction section/ lines 96: https://doi.org/10.1039/D0CC02569K
9- How do the authors explain the different swelling ratios in PBS vs DW?
Author Response
Reviewer 3
The authors have submitted an interesting article " Mussel-inspired surface functionalization of porous albumin gels supporting synergistic antibacterial/antioxidant activity and bone-like apatite formation" which deals with the implementation of mussel-inspired chemistry to boost the antimicrobial property of porous albumin gels, as well as promoting mineralization of in bone. The manuscript is well structured and reads well overall, although it will need some revisions. I suggest this article be published after a major revision.
Thank you for giving us the chance to revise the manuscript. Based on your comments and suggestions, we have made careful modifications to the original manuscript and carefully proofread the manuscript to minimize typographical and grammatical errors. Herein, we provide the responses (as italicized text) to the reviewers’ comments (in bold). Moreover, a revision has been made in the article according to the reviewers’ comments, and changes are highlighted within the revised version of the manuscript.
*** General comments:
ü The abstract is clear and comprehensive, and it comprises all cornerstones including a brief/general introduction to the topic, a non-technical summary of the major findings, and their implications.
ü The introduction is compelling, clear, and concise. The introduction part covers a proper description of the challenge/gap and a strong background in the field associated with a fair literature review.
ü The various sections of the body of the text are clear and concise overall.
ü The conclusions are logical.
*** Suggested revisions:
- First of all, I strongly recommend the authors provide a simple, high-quality, and informative “Graphical abstract” which can present the whole concept of your study at a glance. I would like to recommend authors design a “Graphical Abstract” for this study to better show the whole story in a simple and informative manner. In this regard, you can use illustrate a simple sketch of the big picture and add elements like Schematic1 along with some SEM images along with Live/Dead and a simple schematic of grafting maybe (totally up to you)
Response: We are thankful to the reviewer for your detailed suggestions; here is the graphical abstract of our manuscript in accordance with your kind remarks.
- Please carefully revise the manuscript to remove grammatical errors and vague sentences. Some of the sentences are unnecessarily long which makes it difficult and boring for the readers to follow them. Please double-check the whole manuscript and revise all.
Response: Thank you for your comment. We have carefully proofread the manuscript to minimize typographical and grammatical errors
* I strongly suggest the author keep consistency in the figure's presenting style. For example, FTIR spectra presented in the main body of the manuscript is in Transmittance but supplementary material is presented based on the absorbance rate!
Response: Thank you for noticing this; we have revised the figure’s presenting style (transmittance for all FTIR figures) to keep the consistency.
* Moreover, the scale bare presenting style in figure 3 is different from the rest of the manuscript.
Response: Thank you for your comment. We have revised the scale bar of Figure 3 to keep the consistency with the rest of the figures of the manuscript.
* Furthermore, it is better to be consistent in the way of labeling section of the figure (from left to the right or top to bottom – see figure 4 vs figure 5). Besides, Please decrease the font size of writings in some figures such as figures 6, 7, etc.
Response: Thank you for your comment. We have carefully revised the labeling of figures, and also the font size has been decreased.
* I suggest presenting the mass loss profile in an inverted manner (Remaining mass(%)) which is a more intuitive way to present this type of data ( a negative slope - decreasing from 100% ).
Response: We would like to thank the reviewer for giving us the suggestion to present the data of remaining mass (%) in an inverted manner. As advised, the plot has been revised.
- The novelty statement of an article is of significant importance that highlights the importance of the current study and separates it from previously done research. In this work, the novelty statement poorly represents the work and the authors needed more development and better define their hypothesis and objectives and how the presented work differs from already and recently published reports in the field with similar concepts.
- One of the best ways to highlight the outcomes of a study is to tabulate a comparative master table to compare the findings of this study with recently published data from other research groups. I suggest the authors add a such table to the introduction part.
Response: Thank you much for your comment. We agree with the reviewer about the novelty statement of the article; a table (Table 1) has been provided in the revised manuscript to highlight the difference between the current work and the literature.
Table 1 The summary of different preparation methods of BSA-based hydrogels and their finding.
|
Gelation |
Findings |
Limitations |
Mineralization |
Ref. |
|
pH- triggered |
Facile gelation of hydrogels with good biodegradable properties. |
Extreme pH conditions limit the incorporation of cells. |
NA |
[1-3] |
|
Heat-triggered |
Quick, easy, and harmful chemical crosslinkers-free gelation. |
Heat modifies the secondary structure. Cells cannot survive under heat-triggered gelation. |
NA |
[3-6] |
|
Photo-triggered |
Methacrylate albumin (BSAMA) hydrogels presented good cell activities and mineralization, similar to GelMA. |
It entailed an additional step and expense. Incorporated cells might be damaged by UV radiation. Relatively high concentrations (15-25%) of polymer solutions were used. |
BSAMA hydrogels offered a higher quality of mineralization than GelMA.[7] |
[7-12] |
|
Cryogelation
|
A combination of urea and cysteine on BSA led to porous cryogels. |
Gelation-denatured polypeptide chains. |
NA |
[13,14] |
|
BSAMA cryogels displayed a porous, interconnected spongy network with shape recovery. |
Compared to BSAMA alone, the BSAMA-GelMA hybrid exhibited superior cell adhesion. |
NA |
[15] |
|
|
Sequentially mineralizable transparent and opaque porous gels with improved stability. |
They lack multi-functionalities. |
Cryogels exhibited a larger mineral percentage than their hydrogel counterparts. |
[16] |
|
|
For the first time, multifunctional porous albumin gels were fabricated via surface functionalization inspired by mussels, exhibiting synergistic antibacterial/antioxidant activity and enhanced mineralization and cell responses. |
In vivo investigations could be further required. |
Rod-like shaped minerals with a Ca/P ratio of 1.63 (similar to native HA 1.69) were produced by the sequential mineralization of PDA-functionalized BSAMA hydrogels. |
Current study |
- Figure 8. a doesn’t represent proper Live/dead assay data. To avoid confusion, please assign the live images and dead images with green “Live” and red “Dead” at the top left of the images and also provide merged images.
Response: A proper representation of the live/dead assay in accordance with the reviewer’s guidelines has been provided in the revised manuscript (Figure 9) to avoid confusion.
- In the supplementary information, the FTIR spectrum should be revised to Y-axis: Absorbance (a.u.). Same comment for Figure 7.b, and figure 4.a (transmittance (%)) and figure 4.b (Intensity (a.u.)).
Response: Corrections have been made following the reviewer’s comments about the units in the FTIR spectrum. Revision can be seen in the response to comment 2.
- Since the SEM technique is associated with repetitive vacuum conditions (Au/or Pt coating and image-taking steps) which can dry out the microorganism and may change their original shape, therefore, authors are required to explain how they have prepared samples for SEM while preserving the original morphology of these microorganisms.
Response: We agree with the reviewer about the vacuum conditions (Au/or Pt coating and image-taking steps) for SEM analysis, and to avoid confusion, we have deleted the SEM morphology of the cells and have only shown the CLSM morphology data in the revised manuscript (response to comment 5).
- Some of the references in the context are too old (e.g., 1994, 2003, etc.) and it is not acceptable at all. A myriad of research bodies has been published in recent years and you can find similar concepts and cite them in your paper rather than more than 2 or 3 decades old references. Moreover, in the introduction part to better explain the fundamentals of this field, please read and add valuable information from the following key papers as well: Introduction section/ lines 91: “Mussel‐Inspired Biomaterials: From Chemistry to Clinic - https://doi.org/10.1002/btm2.10385 “Introduction section/ lines 94: “Mussel-inspired hydrogels: from design principles to promising applications- https://doi.org/10.1039/C9CS00849G ” Introduction section/ lines 96: https://doi.org/10.1039/D0CC02569K
Response: We have removed the old references, and also added the citations of the key papers in the updated manuscript as advised by the reviewer. Moreover, the revised manuscript includes the research after 2011 only.
- How do the authors explain the different swelling ratios in PBS vs DW?
Response: We have provided an explanation for the swelling rations in PBS vs DW in the revised manuscript.

Round 2
Reviewer 3 Report
The manuscript is well-amended and all my concerns have been covered. I have no further comments.